# Analysis of Microbial Diversity and Community Structure of Rhizosphere Soil of Three *Astragalus* Species Grown in Special High-Cold Environment of Northwestern Yunnan, China

**DOI:** 10.3390/microorganisms12030539

**Published:** 2024-03-07

**Authors:** Jia-Jie Ding, Guo-Jun Zhou, Xiao-Jie Chen, Wei Xu, Xing-Mei Gao, Yong-Zeng Zhang, Bei Jiang, Hai-Feng Li, Kai-Ling Wang

**Affiliations:** 1Yunnan Key Laboratory of Screening and Research on Anti-Pathogenic Plant Resources from Western Yunnan, Dali 671000, China; gansudingjiajie@163.com (J.-J.D.); yaoxuewsw@163.com (G.-J.Z.); cxj20182000@163.com (X.-J.C.); 17716868566@126.com (W.X.); 15758099263@163.com (X.-M.G.); cell_zyz@163.com (Y.-Z.Z.); jiangbei@dali.edu.cn (B.J.); 2Institute of Materia Medica, College of Pharmacy, Dali University, Dali 671000, China

**Keywords:** *Astragalus*, rhizosphere microbiome, special high-cold environment, northwestern Yunnan, China, metagenomics

## Abstract

*Astragalus* is a medicinal plant with obvious rhizosphere effects. At present, there are many *Astragalus* plants with high application value but low recognition and resource reserves in the northwestern area of Yunnan province, China. In this study, metagenomics was used to analyze the microbial diversity and community structure of rhizosphere soil of *A. forrestii*, *A. acaulis*, and *A. ernestii* plants grown in a special high-cold environment of northwestern Yunnan, China, at different altitudes ranging from 3225 to 4353 m. These microbes were taxonomically annotated to obtain 24 phyla and 501 genera for *A. forrestii*, 30 phyla and 504 genera for *A. acaulis*, as well as 39 phyla and 533 genera for *A. ernestii*. Overall, the dominant bacterial phyla included Proteobacteria, Actinobacteria, and Acidobacteria, while the dominant fungal ones were Ascomycota and Basidiomycota. At the genus level, *Bradyrhizobium*, *Afipia*, and *Paraburkholderia* were the most prevalent bacteria, and *Hyaloscypha*, *Pseudogymnoascus*, and *Russula* were the dominant fungal genera. Some of them are considered biocontrol microbes that could sustain the growth and health of host *Astragalus* plants. Redundancy analysis revealed that pH, TN, and SOM had a significant impact on the microbial community structures (*p* < 0.05). Finally, triterpene, flavonoid, polysaccharide, and amino acid metabolisms accounted for a high proportion of the enriched KEGG pathways, which possibly contributed to the synthesis of bioactive constituents in the *Astragalus* plants.

## 1. Introduction

The rhizosphere represents an important interface where plants interact with the soil and its microorganisms [1,2]. When exposed to environmental stress, plants often produce root exudates that are readily released into the soil. These exudates not only alter the physicochemical properties of the rhizosphere soil but also impact the community structure of rhizosphere microorganisms by selectively recruiting specific microbial groups [3,4]. Rhizosphere microorganisms can enhance nutrient absorption and regulate plant growth through processes such as nitrogen fixation, phosphorus dissolution, potassium dissolution, and plant hormone production [5], and protect plants from pathogens by producing antibiotics, lyases, volatiles, and siderophores [5,6]. As a result, plants and their associated microbes often form complex and dynamic relationships that evolve synergistically [5]. The species and abundance of rhizosphere microorganisms vary among different plant species mainly due to the significant influence of climatic factors, soil conditions, and topographic features [7]. Additionally, some metabolites produced by rhizosphere microbes of medicinal plants represent a potential resource for discovering new active ingredients [8].

*Astragalus*, as a group of medicinal plants belonging to the Leguminosae family, are mainly distributed in the high-latitude region, including Inner Mongolia, Shanxi, Gansu, and Heilungkiang provinces of China according to some current records [9]. The dried root of *A. membranaceus* is used to tonify the qi and other functions in traditional Chinese medicine [10]. The *Astragalus* plants exhibit significant rhizosphere effects referred to as differences in soil biophysical–chemical properties and microbial community composition between rhizosphere soils and bulk soils at the same site [11]. Located in a low-latitude area, the Yunnan province of China has similar high-latitude climatic characteristics due to the influences of the plateau environment. During an investigation of the species diversity and exploitation of medicinal resources of distinctive plants and their microorganisms in northwest Yunnan, we found in total about 23 species of *Astragalus* plants, some of which are endemically distributed at altitudes ranging from 2640 to 4479 m. Most of these alpine distributions have rugged mountains and a harsh environment, leading to limited research and utilization of *Astragalus* plants. Therefore, it is necessary and urgent to systematically study the abundant resources of these plants.

In this study, metagenomic sequencing was applied to compare and interpret the diversity and structure of the rhizosphere microbial community between *A. forrestii* (AF), *A. acaulis* (AA), and *A. ernestii* (AE) growing in the special high-cold environment of northwestern Yunnan at altitudes ranging from 3225 to 4353 m. Meanwhile, the physical and chemical properties of the rhizosphere soil were determined using soil chemistry methods. Finally, correlation analysis was performed to examine the relationship between the *Astragalus* microbial communities and soil physicochemical properties. The results suggest that the rhizosphere microorganisms of these three *Astragalus* plants have good diversity and specific species distribution patterns, and many microbes had been proven to be promising sources of bioagents. Herein we speculate that distinct habitat and soil environmental factors should collectively drive the diversity of rhizosphere microbial community structure associated with *Astragalus* plants, and some biocontrol microbial strains might play an important role in the host plant for immune response and surviving against alpine environmental stresses.

## 2. Materials and Methods

### 2.1. Sample Collection

Rhizosphere soil and plant samples were collected in September 2021 at different locations: *Astragalus forrestii*, Xiaozhongdian, Shangri-La, Dêqên Tibetan Autonomous Prefecture, Yunnan province, China (99°49′ E, 27°26′ N; altitude 3225 m), and *Astragalus acaulis* and *Astragalus ernestii*, Baima Snow Mountain, De qin, Dêqên Tibetan Autonomous Prefecture, Yunnan province, China (99°2′ E, 28°21′ N; altitude 4353 m and 4048 m) (Figure 1A and Appendix A). The three species of *Astragalus* growing in different plateau environments (Appendix A) and their plant appearance are shown in Figure 1B. All healthy sample plants were randomly collected at each plot, along with their rhizosphere soil—the soil closely attached to the roots within 2.5 mm [12]. After packing the samples in aseptic bags, they were immediately transported at low temperature to the laboratory, where they were stored at 4 °C until processing. To collect the rhizosphere soil samples, the soil attached to the roots of *Astragalus* was brushed with a sterile brush as previously described [13]. These soil samples were then divided into two parts: One was used for analyzing soil physicochemical properties, and the other was stored at −80 °C until required for soil DNA extraction.

### 2.2. Determination of Soil Physicochemical Properties

The samples were assayed for pH, soil organic matter, total nitrogen, total phosphorus, and total potassium. This study had three groups of test soil samples, and each test was repeated three times. The soil pH was obtained using a pH meter in a soil suspension made from dry soil and deionized water (water:soil = 2.5:1 (*w*:*v*) [4]. The potassium dichromate volumetric method–outside heating method was then used to assay the soil organic matter (SOM) content [14], while the Kjeldahl method was used to determine the total nitrogen (TN) [15]. Similarly, the total phosphorus (TP) content was determined with the NaOH alkali fusion–molybdenum antimony anti-spectrophotometric method [16], while total potassium (TK) was determined with NaHCO_3_.

### 2.3. Genomic DNA Extraction and PCR Amplification

The *Astragalus* rhizosphere soil samples were frozen at −80 °C immediately after collection. After thawing, DNA was extracted from the samples using the CTAB method [17]. Briefly, this involved adding the samples to a lysis solution containing lysozyme and 1000 μL of CTAB buffer, with repeated upside-down mixing to allow sufficient lysis. Total DNA was subsequently extracted with chloroform: isoamyl alcohol (24:1) prior to its precipitation at −20 °C with isopropanol. The precipitate was then washed twice with 75% ethanol and dissolved with ddH_2_O. Finally, 1 μL RNaseA was added to digest the RNA and left at 37 degrees for 15 min. DNA integrity was detected by 0.8% agarose gel electrophoresis, and concentration was measured via NanoDrop 2000 spectrophotometer (Thermo Fisher, Shanghai, China). Amplification and sequencing of the V4 high-variant region of the 16S rRNA gene was determined using a forward primer 515 F (5′-GTGCCAGCMGCCGCGGTAA-3′) and a reverse primer 806 R (5′-GGACTACHVGGGTWTCTAAT-3′) [18]. A forward primer ITS1 F (5′-CTTGGTCATTTAGAGGAAGTAA-3′) and a reverse primer ITS2 R (5′-GCTGCGTTCTTCATCGATGC-3′) were used as fungal amplification primers [19]. The PCR (Bio-Rad, Hercules, CA, USA) system contained PCR Master Mix with GC buffer, 15 μL Phusion Master Mix, 6 μM primer, 10 μL gDNA, and 2 μL H_2_O in a total volume of 30 μL. The PCR reaction steps were conducted with a denaturation step at 98 °C for 1 min, followed by 30 cycles (98 °C, 10 s; 50 °C, 30 s; 72 °C, 30 s), and then with a final extension step at 72 °C for 5 min. The products were detected by electrophoresis using 2% agarose gel, and samples with a bright main strip between 400 and 450 bp were chosen for further experiments.

### 2.4. Illumina MiSeq Sequencing and Processing

Metagenomic sequencing was performed using the Illumina Novaseq high-throughput sequencing platform to obtain raw data of *Astragalus* rhizosphere soil samples. The sequences were uploaded to NCBI SRA under the accession number PRJNA1003623. In order to ensure the reliability of data, the raw sequences were then processed using the Kneaddata 0.7.4 software as follows: After removing the sequencing adapters (parameter: ILLUMINACLIP: adapters_path: 2:30:10), Trimmomatic 0.36 software was used to discard low-quality sequences with average-quality scores of less than 20 using the sliding window trimming method. Similarly, sequences shorter than 50 bp (parameter MINLEN:50) were removed. The resulting data were then subjected to a BLAST search against the host database using the default Bowtie 2.0.0 software to identify and remove host sequences. Finally, FastQC was used for quality control of the final sequences [20].

### 2.5. Bioinformatics and Statistical Analysis

Alpha diversity metrics, including the observed features, Chao1 index, Shannon diversity index, and Simpson index, were determined and compared with the Kruskal–Wallis method. Kraken2 and a self-built microbial database (sequences belonging to bacteria, fungi, archaea, and viruses were screened from NT nucleic acid database and RefSeq whole genome database of the NCBI) were subsequently used to identify the species present in the samples. After calculating the relative abundance of species in the samples with Bracken, Venn diagrams were plotted to determine the number of genera that were common or unique to the different groups. Linear discriminant analysis effect size analysis (LefSe) was then performed to identify differential microorganisms for each group of samples. PCoA analysis was also carried out, based on Bray Curtis distances, to determine the beta diversity. In this case, the closer the samples clustered, the more similar they were in terms of species composition. The co-occurrence network based on the Spearman correlation matrix was used to study the relationship and interaction between bacteria and fungi [21]. The most important interaction was highlighted (top 50 genera), and the Spearman correlation threshold was set to 0.8, *p*  <  0.05. Nodes represent OTUs, and edges connecting nodes represent correlations between OTUs [16]. Eventually, redundancy analysis (RDA) was performed using the VEGAN package in R 4.3.0 to investigate the relationship between rhizosphere microbial structures and soil environmental factors [22].

## 3. Results

### 3.1. Soil Physicochemical Properties

The rhizosphere soil physicochemical properties were significantly different between the three *Astragalus* plants AF, AA, and AE (Table 1), although the pH values were similar, ranging from 5.28 to 5.89. The highest levels of total nitrogen, total phosphorus, and soil organic matter were found in the rhizosphere of AF. However, the amount of total potassium was higher in the soil of AA compared with those of AF and AE.

### 3.2. Differential Microbial Diversity in *Astragalus’s* Rhizosphere

Beta diversity analyses using PERMANOVA (PCoA, Bray–Curtis distance, F = 30.934, *p*-value = 0.003) further confirmed the significant differences between the three groups, with the first (PCoA1) and second (PCoA2) PCoA axes explaining 72.75% and 18.66% of the total variation in the microbial community, respectively (Figure 2A). To gain insight into the microbial diversity and richness of the rhizosphere, the α-diversity (measured by the observed features as well as the Chao1, Shannon, and Simpson diversity indexes) of the three types of *Astragalus* was compared. The results revealed significant differences (Kruskal–Wallis, *p* < 0.05) between the microbial taxonomic groups of the plants, as shown in Table 2. Furthermore, the species accumulation curve (Figure 2B) flattened as the sample size increased, indicating that sequencing nine samples could reflect the microbial community diversity in the soil. The detailed species information of rhizosphere microbial communities of each *Astragalus* plant have been listed at the phylum (Appendix A for the bacterial information and Appendix A for the fungal information of AF, AA, and AE, respectively) and genus levels (Appendix A for the bacterial information and Appendix A for the fungal information of AF, AA, and AE, respectively) in the Appendix A.

### 3.3. Bacterial Community Analysis

The diversity of bacterial communities in the three types of *Astragalus* samples was determined at the phylum (Figure 3A and Appendix A) and genus levels (Figure 3B and Appendix A). Overall, 37 phyla were identified (Appendix A), with Proteobacteria, Actinobacteria, and Acidobacteria being the dominant ones (relative abundance greater than 1%), although their relative abundance actually varied between the samples. In all samples, Proteobacteria was found to be the most abundant phylum, and its proportion ranged from 65.17% for AF to 56.71% for AA and 41.59% for AE (Appendix A). At the genus level, a total of 585 bacterial genera were identified (Appendix A), among which 239 were common to the three *Astragalus* species based on Venn diagram analysis (Figure 4A and Appendix A). *Bradyrhizobium*, *Afipia*, and *Paraburkholderia* were the three most abundant genera in the rhizosphere bacterial communities of the three *Astragalus* species, and *Bradyrhizobium* was the dominant genus, accounting for 49.13%, 29.61%, and 16.66% of the total genera in AF, AA, and AE, respectively (Appendix A). Meanwhile, each bacterial community of *Astragalus rhizosphere* had its own unique genera. Specifically, AF, AA, and AE had 67, 54, and 98 unique genera of rhizosphere bacteria, respectively (Appendix A), and AE had the highest number out of the three groups. Additionally, the proportion of *Afipia* was 1.80%, while that of *Paraburkholderia* was 5.23% in AF, and *Mycobacterium* was found to be several times more abundant in AF compared with the other samples.

### 3.4. Fungal Community Analysis

Compared with the bacterial communities, the fungal ones showed a simpler composition in all samples (Figure 3C,D and Appendix A). For instance, only five fungal phyla, namely Ascomycota, Basidiomycota, Mucoromycota, Chytridiomycota, and Zoopagomycota, were identified (Appendix A), and of these, Ascomycota was the most dominant, accounting for 68.92% (AF), 72.10% (AA), and 67.60% (AE) of the population in the three species of *Astragalus*. Basidiomycota, the second most dominant phylum, was more abundant in AF than in the other samples, while Zoopagomycota was predominantly found in AE but absent from AA. At the genus level, the rhizosphere microbial community of AF, AA, and AE contained 132 (Appendix A), 118 (Appendix A), and 98 (Appendix A) fungal genera, respectively. Over 206 genera, distributed differently across the samples, were identified (Appendix A). Of these, a total of 51, 39, and 23 fungal genera were uniquely distributed in AF, AA, and AE (Appendix A), and 49 fungal genera were commonly owned to the three groups (Figure 4B and Appendix A). The top five genera, in decreasing order of their relative abundance, were *Hyaloscypha*, *Pseudogymnoascus*, *Russula*, *Ilyonectria*, and *Rhizophagus* (Appendix A). More specifically, *Hyaloscypha* accounted for 40.53% (AF), 22.95% (AA), and 2.00% (AF) of the genera in the three species of *Astragalus*, while *Pseudogymnoascus* and *Rhizophagus* accounted for the highest proportion of fungal genera in AE. In addition, *Russula* was mainly present in AF, while *Ilyonectria* was mainly observed in AA and AE. However, it should be noted that, as often observed in studies involving bacterial communities, unclassified genera accounted for 23.85% of the population, and being a non-negligible proportion, they could potentially be the focus of future studies.

### 3.5. Identification of Microbial Biomarkers

LEfSe analysis was performed on the taxonomical compositions of rhizosphere microbial communities to identify biomarkers for each *Astragalus* species, with an LDA score above 3.5 selected as the threshold for differential analysis. A total of 32 significantly enriched bacterial taxa were found in the three groups of samples (*p* < 0.05) (Figure 5A), of which 21 were present in AE. At the phylum level, Proteobacteria, Actinobacteria, Firmicutes, and Cyanobacteria were identified as the biomarkers for AA and AE, while the orders Hyphomicrobiales and Rhodospirillales were identified as the biomarkers for AF and AA, respectively (Figure 5B). In the case of fungi, 49 fungal taxa were significantly enriched in the three samples, with 19 being differentially enriched in AF. In addition, for AA, the differentially abundant taxa included the genus *Cenococcum*, the family Gloniaceae, and the order Cantharellales, while for AE, the order Hypocreales, the family Nectriaceae, and genus *Beauveria* were differentially abundant.

### 3.6. Relationship between Microbial Communities and Soil Physicochemical Properties

To explore the relationship between microbial community structures and soil environmental factors, RDA (redundancy analysis) was first performed for the top 20 bacterial genera. The results showed that RDA1 and RDA2 could explain 59.55% and 26.12% of the total variation, respectively (Figure 6A). Correlation analyses further revealed that *Bradyrhizobium*, *Paraburkholderia*, and *Mycobacterium* were positively correlated with TN, TP, and SOM but negatively correlated with pH and TK. In contrast, *Afipia* showed a strong positive correlation with pH and TK. The above analyses were also performed for the top 20 fungal genera. In this case, the first two axes (RDA1 and RDA2) explained 56.64% and 26.9% of the total variation, respectively (Figure 6B). *Hyaloscypha* and *Russula* were both positively correlated with TN, TP, and SOM but negatively correlated with pH and TK. On the other hand, *Pseudogymnoascus* and *Rhizophagus* had a significantly positive correlation with pH but a negative one with the other indices.

### 3.7. KEGG Pathway Enrichment Analysis

The metabolic pathways that could be involved in the rhizosphere soil microorganisms of the three *Astragalus* species were assessed by KEGG annotation, with metabolism being the most enriched (Figure 7 and Appendix A). In particular, the most significant pathways were amino acid metabolism, carbohydrate metabolism, terpenoid and polyketide metabolism, and the biosynthesis of other secondary metabolites, such as flavone and flavonol biosynthesis, lipopolysaccharide biosynthesis, as well as sesquiterpenoid and triterpenoid biosynthesis. Other relevant metabolic pathways are also listed in Appendix A.

### 3.8. Co-Occurrence Network Analysis of Microbial Community

Co-occurrence network analysis was used to evaluate the complexity of the interactions between microorganisms detected in the rhizosphere of three *Astragalus* species (top 50 genera) (Figure 8 and Appendix A). The networks consisted of 43 nodes and 119 edges. These genera mainly belong to Proteobacteria and Actinobacteria, with most belonging to Proteobacteria. The top genera, *Bradyrhizobium*, were identified as hub biomarkers in the rhizosphere due to their high abundance. These microbial communities constituted 62 positive correlations and 57 negative correlations. Among them, *Bradyrhizobium* was significantly positively correlated with *Paraburkholderia* and negatively correlated with *Variovorax*.

## 4. Discussion

In this study, the predominant bacterial communities were composed of Proteobacteria, Actinobacteria, and Acidobacteria (Appendix A). This observation was, in fact, aligned with the general idea that these phyla tend to be common in soils worldwide [23]. Furthermore, the fragile and challenging habitat of *A. ernestii* plants showed the highest percentage of Actinobacteria, and this was consistent with Zhang et al.’s study [24], which found that the relative abundance of Actinobacteria was higher in oligotrophic environments compared with copiotrophic ones. Members of this phylum are known for their ability to produce extracellular hydrolytic enzymes that decompose resilient organic matter such as cellulose and chitin, as well as animal and plant residues. These processes, in turn, not only allow the microorganisms to thrive in oligotrophic environments [25] but also enhance soil carbon cycling [24,26]. In addition to being prolific producers of antibiotics [7,27], Actinobacteria can also produce structurally complex secondary metabolites that possess diverse biological activities [28,29,30], with many of these finding applications in medicine and agriculture [31,32,33]. Moreover, members of this phylum can utilize carbohydrates [34] and are known to attach to the rhizosphere surface, produce polysaccharides that promote root adhesion to soil particles and protect the roots [35]. Proteobacteria and Chloroflexi have been considered the major producers of natural phenolic and aromatic compounds in soil [26,36]. Some genomic studies suggested that many of their gene sequences should be related to the nitrogen fixation and carbon cycling of these bacteria [35].

As far as the fungal communities were concerned, the dominant phyla in all samples were Ascomycota and Basidiomycota (Appendix A), which is consistent with the findings of soil fungi in terrestrial ecosystems [37] and the results of studies on the *Stipa purpurea* of the Qinghai–Tibet Plateau [13]. The ability to degrade cellulose and lignins is widespread in fungi derived from soil [38], and is particularly well represented in Ascomycota and Basidiomycota [39]. Basidiomycota is positively correlated with soil C and N [40], and a large proportion of this phylum can actually form mycorrhizas with plant roots in soil [38]. Ascomycota is more prevalent in harsh habitats, while Basidiomycota tends to thrive in nutrient-rich environments [26], hence justifying its highest abundance in AF. Furthermore, the diversity and composition of the fungal communities in the three *Astragalus* rhizospheres are significantly influenced by the altitude. In particular, Ascomycota was more dominant in AA at high altitude (4353 m). This finding was also mentioned in previous research that reported the microbial shift from Basidiomycota to Ascomycota with increasing elevation [41]. However, there are some things that can not be neglected, that is, a high percentage of unclassified microorganisms was observed in both bacterial and fungal microbiomes, which means that this special living environment may breed many unknown microbial species for further research using multiple high throughput sequencing and culturable approaches. During the investigation of microbial species diversity derived from *Astragalus* plants, our group had isolated a new fungal species *Varicosporellopsis shangrilaensis* from the rhizosphere soil of *A. polycladus* [42]. The new species, as the third species of the genus *Varicosporellopsis*, was firstly discovered in a high-altitude, cold, terrestrial environment.

A decrease in biodiversity with increasing altitude is one of the most common phenomena observed in community ecology [37], but the current results were inconsistent with that trend. Specifically, 369, 386, and 435 bacteria as well as 132, 118, and 98 fungi were respectively detected in AF, AA, and AE (Appendix A). This may be due to the ecological differences of the growing environment of these three *Astragalus* plants. The *A. ernestii* plants grow in a harsher environment, characterized by more sand and gravel (Appendix A). It needs to recruit more microbial communities that specifically assist in overcoming environmental stresses. In contrast, the plants of *A. forrestii* and *A. acaulis* grow in wetter meadows that are richer in nutrients (Appendix A), and this is reflected in the higher abundance of fungi. Thus, it is likely that altitude represents only one of the reasons affecting species diversity. Indeed, soil characteristics, plant species, root exudates, and the period of plant growth are some other factors that also significantly affect the composition of the rhizosphere microbial community [43]. *Bradyrhizobium* was the dominant bacterial genus and accounted for 16.66% (AE), 29.61% (AA), and 49.13% (AF), respectively. Members of this genus are mainly present in the roots of the leguminous plant. They convert nitrogen from the air into nitrogenous substances that can be absorbed by the plant [44]. Therefore, it is not surprising that the genus *Bradyrhizobium* was detected in large numbers in all the three species of the leguminous *Astragalus* plants. The rhizosphere soil of AF, which has the highest nitrogen content of the three *Astragalus* species, had the highest abundance of *Bradyrhizobium*. Besides this, some bacteria in the commonly dominant genera displayed biological control functions for plant disease. For example, *Pseudomonas* can produce the antifungal compound 2,4-diacetylphloroglucinol [5], while *Variovorax paradoxus* can help *Panax notoginseng* resist root rot disease [45]. Similarly, *Penicillium* can produce antibiotic-like substances that protect plants from attacks by pathogens [4]. Among the top 20 dominant fungal genera (Appendix A), the genus *Russula* was only detected in AF, while *Cladosporium*, *Microthyrium*, and *Cladobotryum* were only detected in AA. It is reported that some members of *Russula* are important ectomycorrhizal fungi that play a key role in soil nutrient uptake and cycling [46]. Also, the genus *Cladobotryum* has been considered a great potential antifungal resource against fungal phytopathogens [47], and its secondary metabolites displayed chemical and bioactive diversity [48]. Above all, we found that different *Astragalus* varieties recruit various microorganisms to form specific rhizosphere microbial colonies to maintain the growth and health of host plants.

Some beneficial rhizosphere microbes and their secondary metabolites might be very important for bio-controlling pathogens of *Astragalus* plants. A growing body of research has shown that some active medicinal ingredients of plants are actually produced by their rhizosphere microorganisms [49]. At the same time, it has been reported that harsh ecological conditions, such as low temperatures and high ultraviolet radiation, that prevail in the alpine region can result in unique metabolic profiles for microorganisms [50]. In this study, amino acid, terpenoid, and polyketide metabolism were annotated to KEGG, which means that some rhizosphere microbes derived from *Astragalus* plants could produce such flavonoids and flavonols, lipopolysaccharides, and terpenoids in much higher yields. According to Abd Elrahim Abd Elkader et al. [10], triterpenes, flavonoids, polysaccharides, and amino acids have been identified as the primary functional components of *Astragalus*. Indeed, in a previous study, flavonoids and terpenoids were isolated from the roots of *A. ernestii*, which have been used as a substitute for *Radix Astragali* (Huang-Qi) in southwestern China [9]. These findings further suggest that the secondary metabolites of these rhizosphere microorganisms should play a crucial role in synthesizing the main active natural components of *Astragalus* plants, and some microbes may be developed and utilized as a great potential medicine resource.

In this study, RDA correlation analysis was performed to determine the relationship between the structure of microbial communities at the genus level and soil environmental factors. As shown in Figure 6, the genera *Afipia*, *Variovorax*, *Pseudomonas*, and *Acidiphilium* showed a positive correlation with pH and TK but a negative correlation with TN, TP, and SOM. These findings demonstrate that higher levels of pH and TK should have contributed to the abundance of these microbes, while the other soil factors had the opposite effects. Several studies have reported that the diversity of fungal communities was closely related to potassium levels. In addition, soil bacteria also influence the solubility and effectiveness of potassium, which, in turn, favor the selection of specific bacteria associated with potassium concentration [31]. The dominant genera *Bradyrhizobium* and *Hyaloscypha* were positively correlated with TP, TN, and SOM. This is also associated with the fact that the highest TP, TN, and SOM contents in AF are related to the high proportion of these two genera in it. pH is considered to be one of the most important soil parameters [51]. Soil pH affects the solubility of nutrients and metabolism of microorganisms [52,53]. High rainfall and the decomposition of organic matter can enhance soil acidity [54,55]. Vegetation density and apomixis are higher around AF distributed at low altitude, and the soil has a higher content of organic matter and humus. The acidic soil environment recruits acidophilic bacteria such as *Streptophilus* sp. to accumulate in the soil. Then they participate in the nitrogen cycle, decomposition of organic matter, and provision of nutrients to the soil. The amount of nitrate and available phosphorus in rhizosphere soil tends to decrease with elevation [56]. At the same time, phosphorus, which is the second most important nutrient for plant growth after nitrogen, is mostly present in insoluble form in the soil, and as such, it cannot be directly utilized by plants [57]. In this case, soil or root microorganisms, such as *Bacillus*, *Burkholderia*, *Pseudomonas*, and *Penicillium*, can solubilize the combined phosphorus in soil into a water-soluble one that can be absorbed by plants [58].

## 5. Conclusions

In this study, we explored for the first time the rhizosphere microbial structures of *A. forrestii*, *A. acaulis*, and *A. ernestii* plants grown in a special high-altitude, cold environment of northwestern Yunnan, China, which had a rich diversity and certain novelty. Many biocontrol and growth-promoting microorganisms may play an important role in helping *Astragalus* plants to grow healthy and enhance their resilience and adaptability in extreme environments. Soil physicochemical properties, such as pH, TN, and SOM, were identified as the main driving factors affecting the composition and structure of the communities of the three groups. This study is of practical significance for guiding the protection and utilization of *Astragalus* plants as well as their microbial biological resources in special environments.

## Figures and Tables

**Figure 1 microorganisms-12-00539-f001:**
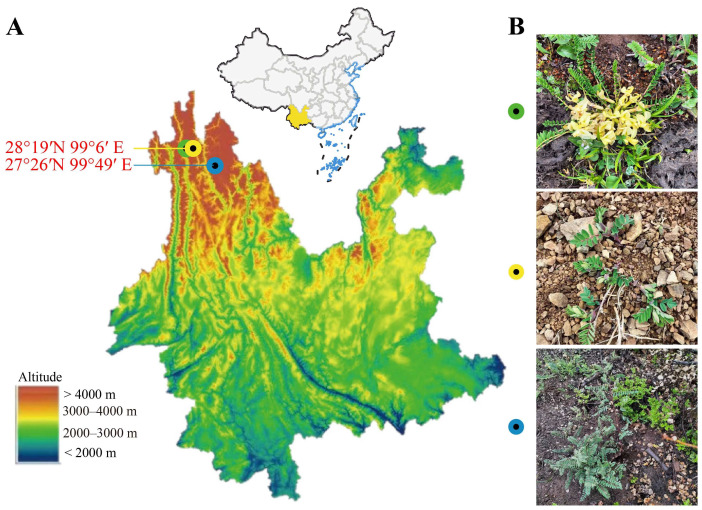
Map of sampling sites in the northwestern area of Yunnan province, China (**A**). In the picture (**B**), the green dot is the sampling site for *Astragalus acaulis* growing in high-altitude grassland at 4353 m; as shown in the yellow dot, *A. ernestii* grows in the steep slopes of flowstone beaches at 4048 m; and the blue dot shows the *A. forrestii* survival environment, surrounded by more pine trees and deciduous foliage at 3225 m.

**Figure 2 microorganisms-12-00539-f002:**
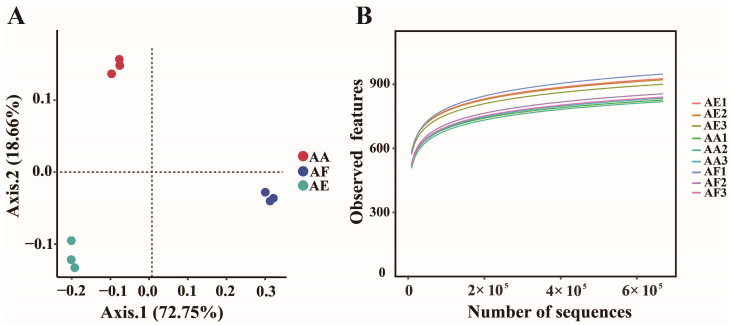
Principal coordinate analysis (PCoA) (**A**) and rarefaction curves of nine rhizosphere soil samples (**B**) of *Astragalus forrestii* (AF), *A. acaulis* (AA), and *A. ernestii* (AE).

**Figure 3 microorganisms-12-00539-f003:**
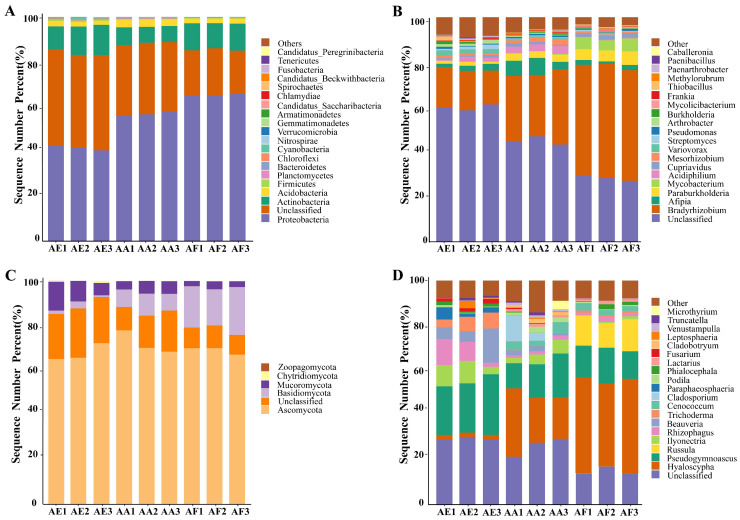
Taxonomical composition of rhizosphere microbial communities in the three *Astragalus* species of *A. forrestii* (AF), *A. acaulis* (AA), and *A. ernestii* (AE) at different taxonomic levels. The relative abundance of bacteria at the phylum (**A**) and genus (**B**) levels, and fungi at the phylum (**C**) and genus (**D**) levels.

**Figure 4 microorganisms-12-00539-f004:**
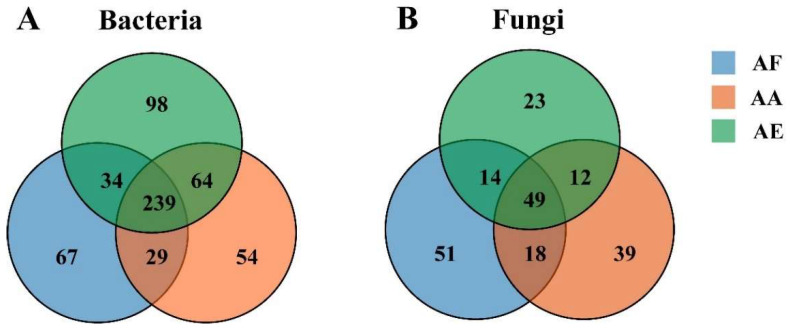
Venn diagram showing the number of bacteria (**A**) and fungi (**B**) at the genus level in the rhizosphere soil samples of *Astragalus forrestii* (AF), *A. acaulis* (AA), and *A. ernestii* (AE). Each circle, with a different color in the diagram, represents the number of species specific to the corresponding subgroup. Middle core numbers represent the number of genera common to all groups.

**Figure 5 microorganisms-12-00539-f005:**
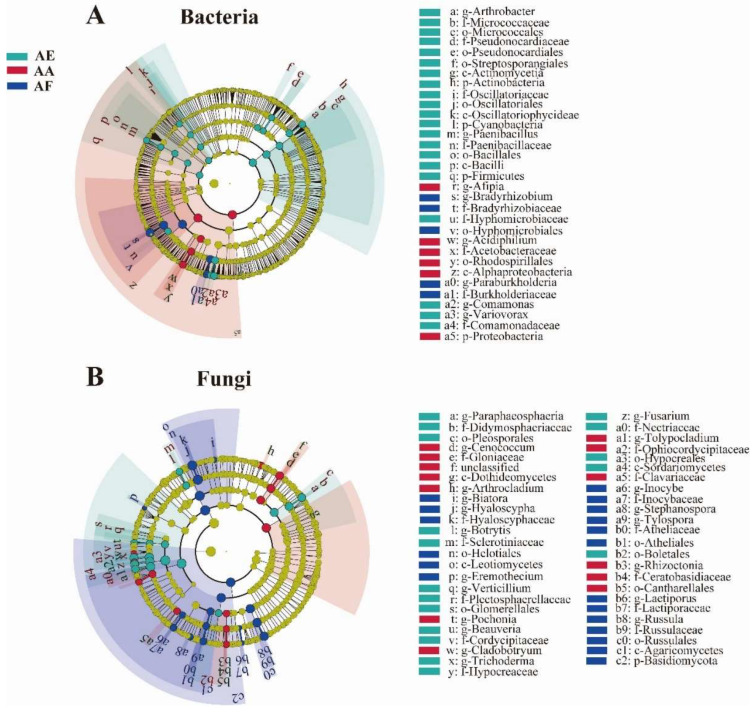
LEfSe analysis shows the different bacterial (**A**) and fungal (**B**) biomarkers in the rhizosphere of *Astragalus forrestii* (AF), *A. acaulis* (AA), and *A. ernestii* (AE). Differently colored regions represent different constituents, with the diameter of each circle being proportional to the relative abundance of the taxon. The inner to outer circle corresponds to the taxonomic levels, from the phylum to the genus.

**Figure 6 microorganisms-12-00539-f006:**
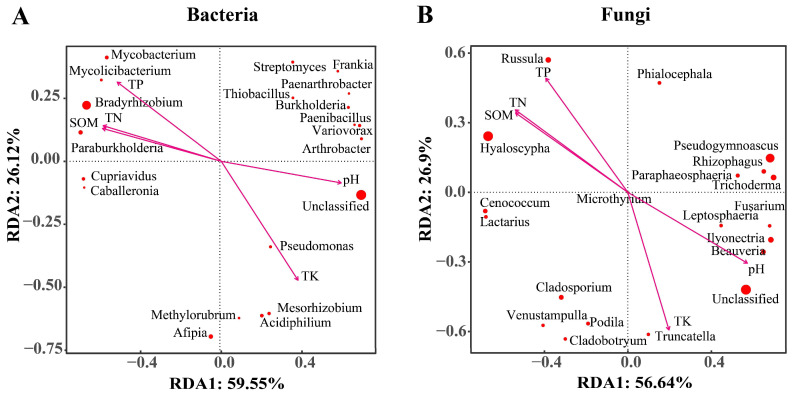
Correlations between soil physicochemical properties and the community structure for bacteria (**A**) and fungi (**B**), as determined by redundancy analysis at the genus level. The environmental factors are represented by arrows; each point represents a genus, and the larger the point, the higher the abundance of the corresponding genus.

**Figure 7 microorganisms-12-00539-f007:**
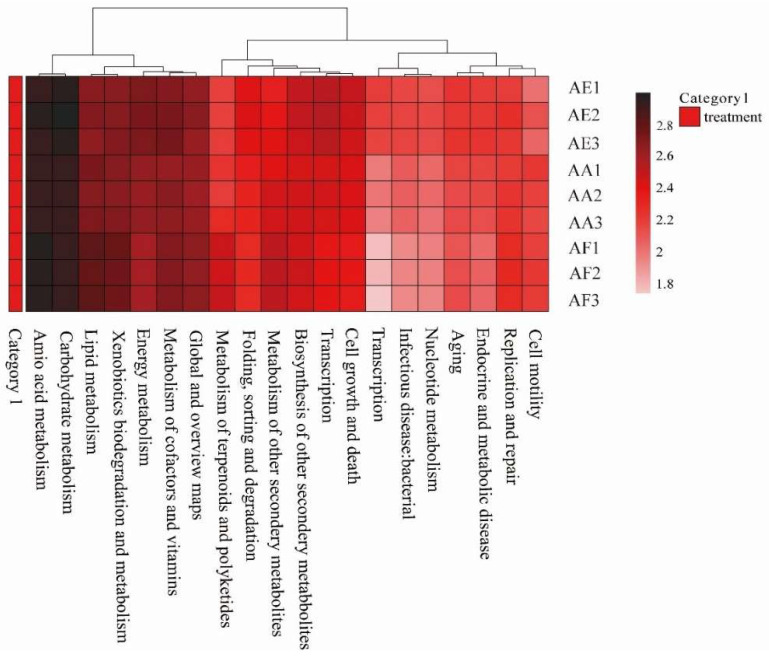
Heat map showing the abundance of level 2 KEGG metabolic pathways of the rhizosphere microorganisms of *Astragalus forrestii* (AF), *A. acaulis* (AA), and *A. ernestii* (AE).

**Figure 8 microorganisms-12-00539-f008:**
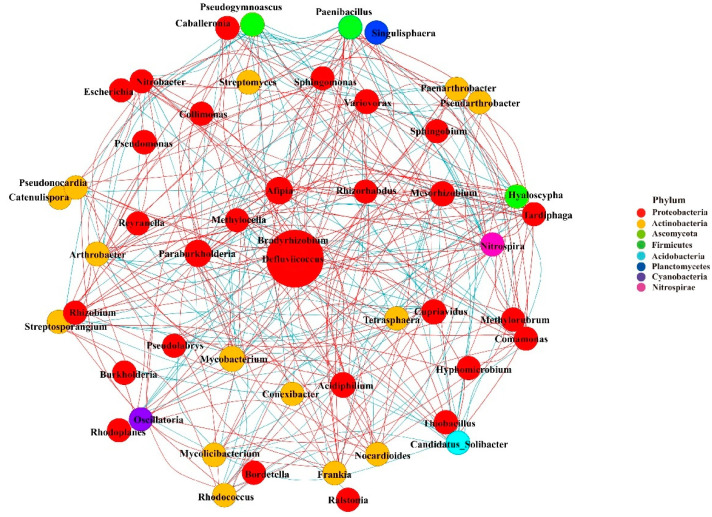
Co-occurrence network analysis of rhizosphere microbial communities of *Astragalus forrestii* (AF), *A. acaulis* (AA), and *A. ernestii* (AE) (top 50 genera). The size of each node (representing OTUs) is proportional to the number of connections (degrees). The size of edges connecting nodes represent both strong (Spearman’s ρ > 0.80) and significant (*p* ≤ 0.05) correlations between OTUs. Node colors represent the taxa indicated.

**Table 1 microorganisms-12-00539-t001:** Soil physicochemical properties of *Astragalus forrestii* (AF), *A. acaulis* (AA), and *A. ernestii* (AE).

Sample ID	pH	TN (g/kg)	TP (g/kg)	TK (g/kg)	SOM (g/kg)
AF	5.29 ± 0.01 a	3.37 ± 0.02 a	1.66 ± 0.01	13.04 ± 0.09 a	55.12 ± 0.79 a
AA	5.60 ± 0.01 a	2.65 ± 0.02 a	1.00 ± 0.02	18.33 ± 0.10 a	46.32 ± 0.03 a
AE	5.88 ± 0.02 a	2.18 ± 0.02 a	0.93 ± 0.04	16.98 ± 0.06 a	40.12 ± 0.13 a

SOM, soil organic matter; TN, total nitrogen; TP, total phosphorus; TK, total potassium. AF, *Astragalus forrestii*; AE, *Astragalus acaulis*; AE, *Astragalus ernestii*. Data shown represent the average of three replicates and their standard deviations. Lowercase letter a indicates significant differences (*p* < 0.05) based on the Kruskal–Wallis test.

**Table 2 microorganisms-12-00539-t002:** Alpha diversity metrics of bacteria and fungi in the rhizosphere soil of *Astragalus forrestii* (AF), *A. acaulis* (AA), and *A. ernestii* (AE).

Sample ID	Chao1	Observed Features	Shannon	Simpson
AF	1044.76 ± 98.81 a	962.67 ± 82.25 a	4.52 ± 0.11 a	0.86 ± 0.01 a
AA	908.19 ± 8.32 a	855.67 ± 11.59 a	4.82 ± 0.04 a	0.88 ± 0.01 a
AE	1020.6 ± 8.98 a	946.67 ± 14.74 a	4.66 ± 0.09 a	0.82 ± 0.01 a

The observed features index was used to evaluate the number of observable OTUs; the Chao 1 and ACE indices were used to evaluate species richness; the Shannon and Simpson indices were used to evaluate species diversity. Data shown represent the average of three replicates and their standard deviations. The lowercase letter a indicates significant differences (*p* < 0.05) based on the Kruskal–Wallis test.

## Data Availability

The data presented in this study are deposited in the NCBI repository, with accession number PRJNA1003623.

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
