# Peer review of "Analysis of Microbial Diversity and Community Structure of Rhizosphere Soil of Three Astragalus Species Grown in Special High-Cold Environment of Northwestern Yunnan, China"

_microorganisms, 2024, doi:10.3390/microorganisms12030539_

Round 1

Reviewer 1 Report

Comments and Suggestions for Authors

The article “Analysis of microbial diversity and community structure of rhizosphere soil of three Astragalus species grown in special high-cold environment of northwestern Yunnan, China”, analyzed the microbial diversity and community structure of rhizosphere soil of A. forrestii, A. acaulis, and A. ernestii plants grown in special high-cold environment. It comprehensively explains the rhizosphere microorganisms associated with Astragalus species in special high-cold environments. I believe it has merit to be published in the current journal. However, I have a few comments to improve it.

Introduction: It is well written; I feel the hypothesis is missing at the end of the introduction.

Material and Methods:

Why was Kruskal–Wallis method used? Non-parametric tests are usually almost as powerful as parametric tests in circumstances where the parametric tests are appropriate. However, it is necessary to show the reason for using this method.

Results and Discussion:

Lines 423-452 and 453 – 516: Paragraphs are too long.

Author Response

Dear reviewer,

Thank you so much for reviewing our manuscript. Your comments immensely helped us to improve the manuscript. The manuscript has been revised according to your suggestions. All the comments and suggestions have been replied point-by-point as follows:

Point 1

Location: Introduction

Comment: It is well written; I feel the hypothesis is missing at the end of the introduction.

Response: Thanks very much for your suggestion. The hypothesis has been added at the end of the introduction. Please see the lines 96-101.

Point 2

Location: Material and Methods

Response: Thanks very much for your suggestion. Kruskal-Wallis is one of the non-parametric tests, because this test does not involve the parameters of the overall distribution in the process of inferring the overall from the sample data, does not require the overall distribution to satisfy certain conditions, and the sample size does not have to be very large, therefore this method was chosen.

Point 3

Location: Results and Discussion

Comment: Lines 423-452 and 453-516: Paragraphs are too long.

Response: Thanks very much for your suggestion. To provide clear guidance for the readers, the relevant content of lines 423-452 and 453-516 have been divided into two paragraphs in the revised manuscript. Please see the lines 455-496 (the 3rd paragraph) and 497-518 (the 4th paragraph) in the revised Discussion section.

Reviewer 2 Report

Comments and Suggestions for Authors

In the introduction several sentences appear more related to results or conclusion.

In material and methods section there is only evidence of the amplification of 16S universal target for bacteria. The way you obtained data from fungi is totally missed.

Comments on the Quality of English Language

The overall quality of the presentation is suitable to support the comprehension of the work, however, some parts of the text need a moderate revision.

Author Response

Point1

Location: Introduction

Comment: In the introduction several sentences appear more related to results or conclusion.

Response: Thanks very much for your suggestion. I have deleted several sentences appearing more related to results or conclusions in the introduction. Please see the revised introduction.

Point2

Location: Material and Methods

Comment: In material and methods section there is only evidence of the amplification of 16S universal target for bacteria. The way you obtained data from fungi is totally missed.

Response: Thanks very much for your suggestion. We have added the data of fungal primers and related reference in the revised manuscript. Please see the lines 151-154 and 682-684.
